# Comparison of Physicians’ Attitudes and Practice Regarding Vaccination during Pregnancy in Turkey

**DOI:** 10.3390/vaccines12070798

**Published:** 2024-07-18

**Authors:** Ateş Kara, Hasan Tezer, Ergin Çiftçi, İhsan Ateş

**Affiliations:** 1Department of Pediatrics, Pediatric Infectious Disease Unit, Hacettepe University Faculty of Medicine, Health Institutes of Turkey, Turkish Vaccine Institute, 06100 Ankara, Turkey; 2Department of Pediatrics, Pediatric Infectious Disease Unit, Gazi University Faculty of Medicine, 06560 Ankara, Turkey; hasantezer@yahoo.com; 3Department of Pediatrics, Pediatric Infectious Disease Unit, Ankara University Faculty of Medicine, 06100 Ankara, Turkey; erginciftci.dernek@gmail.com; 4Department of Internal Medicine, Health Science University, Ankara City Hospital, 06800 Ankara, Turkey

**Keywords:** pregnancy, vaccination, maternal immunization, vaccines, placental transport, infant disease

## Abstract

This study aimed to investigate the knowledge, attitudes, and behaviors of family physicians (FPs), pediatricians (PPs), and obstetricians and gynecologists (OGs) regarding vaccine administration during pregnancy in Turkey as factors that contribute to decision-making. The survey was distributed among FPs, OGs, and PPs, and participants were asked to rate their knowledge on specific topics using a five-point scale ranging from “Not Effective” to “Effective”. The tetanus and diphtheria (Td) vaccine was highly recommended by 94.9% of physicians and considered very effective. Among the physicians surveyed, 80% of PPs and 66.0% of OGs were aware of the disease burden of pertussis. We also found that 74.5% of FPs and 77.2% of PPs believed they needed more information about vaccination during pregnancy. All physicians surveyed agreed or strongly agreed that explaining the disease risks and benefits of vaccination increases the vaccine acceptance rate. The results of this survey study indicate that the knowledge and awareness of physicians need to be improved to increase vaccination rates during pregnancy in Turkey, and it is essential to incorporate influenza and tetanus, diphtheria, and pertussis (TdaP) vaccines into the standard maternal immunization schedule for newborns.

## 1. Introduction

Immunization during pregnancy is an important prevention strategy with a single intervention. Mothers and, most importantly, infants benefit from a safe and efficient immunization program. Hormonal changes taking place during pregnancy and early infancy may alter the immune system response, leaving mothers and infants vulnerable to infections that can be prevented by immunization. Several studies have shown that a decrease in immune system response during pregnancy does not affect the effectiveness of vaccination. Thus, immunization during pregnancy provides both direct and indirect protection to pregnant women, especially during the third trimester and postpartum periods. Maternal vaccines have the potential to provide clinically significant protection for mothers and infants. This is achieved via antibodies induced by vaccination during pregnancy, and these antibodies protect infants from birth until the primary immunization series is initiated and provide significant protection for newborns from many diseases, like tetanus and whooping cough, which can be fatal [1,2].

Although various healthcare organizations worldwide have issued specific guidelines for vaccinating pregnant women, there are still concerns among women and physicians about which vaccines should be administered during pregnancy [3]. Maternal immunization, despite its proven safety and efficacy, is underused and not widely accepted, even in developed countries. There are many reasons for poor compliance with maternal immunization recommendations. The barriers to the uptake of vaccines by pregnant women are more complex than those observed for low uptake in infant vaccination programs [4]. The focus group is healthy adult mothers who might not realize the importance of vaccinations throughout their lives [5]. There is also a surge of information on social media urging expectant mothers to steer clear of possibly dangerous substances. Furthermore, the promotion of vaccines for mothers is primarily driven by their family physicians (FPs), pediatricians (PPs), and obstetricians and gynecologists (OGs) [5,6,7]. It is possible that some physicians are not fully aware of the severe implications of diseases that vaccines can prevent in mothers and infants. Additionally, the economic burden on an expectant mother and her family is one of the primary reasons for low immunization rates. However, the NIS data found that the main reason for not being vaccinated was vaccine safety or efficacy (44.8%), and cost was only 5.6% [8]. The success of maternal vaccination is demonstrated by the maternal and neonatal tetanus elimination program in reducing the burden of neonatal tetanus, especially in developing countries [9]. Therefore, the recommendation and uptake rate of the Td vaccine has increased due to its proven efficacy and, most importantly, safety. After the Td vaccine success story, influenza and pertussis vaccines for the prevention of influenza and pertussis, which have the potential to increase mortality and morbidity rates in mothers and infants, were integrated into the maternal vaccination guidelines in many countries. Many publications show these vaccines’ effectiveness and safety in mothers and infants [10,11,12,13,14,15]. However, despite the proven benefits of influenza and tetanus, diphtheria, and pertussis (TdaP) vaccines, the uptake of these vaccines is still disappointing, even in developed countries [5,11,12,13].

In order to understand the reasons behind this low uptake in Turkey, we investigated the knowledge, attitudes, and behaviors of FPs, PPs, and OG physicians toward influenza, tetanus, and diphtheria (Td) and TdaP vaccination during pregnancy as factors that influence these physicians in their vaccination decisions.

## 2. Materials and Methods

This survey was planned as a self-answered questionnaire, and a draft questionnaire was distributed on 18 December 2021 to a selected group of FPs, OGs, and PPs seeking comments and evaluation. After the initial opinions were collected, the questionnaire was finalized. After this preparatory phase, an Ethics Committee application was made, and the study was approved by the Local Ethics Committee of Ankara City Hospital, with the registration number and date of 31 January 2022/decision No.: E2-22-2518. After ethics approval, the final questionnaire was distributed among FPs, OGs, and PPs on 7 February 2022 via e-mail, with the questionnaire as an attachment, following a notification letter sent by the coordinating committee for the study. The study was conducted and completed during the initial phase of the COVID-19 pandemic, and 2387 physicians from different hospitals and cities in Turkey responded. Answered questionnaires were entered into a database utilizing a double-data entry method, and the investigators cleaned the database. This study was conducted by strictly observing the Declaration of Helsinki, and informed consent was obtained from all participants who provided answers to the questionnaires.

The survey included multiple-choice, closed-ended, and semi-closed-ended questions (Appendix A). The participants selected only one answer to the questions regarding their knowledge about the questions using a five-score test, where five is “Effective”, and one is “Not Effective”. The total survey score for every question was calculated by summing the scores obtained through this coding. The investigating team exerted efforts to reach respondent physicians to complete the missing values, and various queries were generated to provide a complete database due to the low number of respondents to the questionnaires sent by email. As the response rates were not followed using a web-based system, no IP or cookie checks were performed, and no log files were created. Additionally, no statistical corrections were performed.

Statistical Analysis: Pearson’s chi-square test was used to compare the categorical variables, the Mann–Whitney U test was used for two independent samples, and the Kruskal– Wallis test was used for multiple independent samples to determine the factors predicting vaccine recommendations during pregnancy. Statistical significance was defined as *p* < 0.05. Missing values, which were not completed through queries, were not extrapolated. 

## 3. Results

### 3.1. Study Population

The study population consisted of practicing family physicians, pediatricians, and obstetricians/gynecologists randomly selected from their respective national society databases. The survey data were entered using the double-data entry method, and only surveys with an 80% completion rate as a minimum were entered into the database. Family physicians had the highest response rate (54.3% of all responses received), followed by PPs (29.9%) and OGs with the lowest response rate (15.8%) (Table 1). 

### 3.2. Physician Perceptions

The most commonly recommended vaccine was Td, reaching 98,1% (94.9% recommended routinely plus 3.2% recommended in case of risk) of all participating physicians. Almost half of the physicians (45.5%) recommended the TdaP vaccine as part of their routine practice during pregnancy, with OGs having the highest recommendation rate at 76.6%. The study found that a similar proportion of physicians routinely advised their pregnant patients to get vaccinated for influenza (Table 2). 

However, only 39.5% of all physicians considered the influenza vaccine to be “very effective” during pregnancy, while all OG physicians rated it as “effective”. Similarly, OG physicians believed that the TdaP vaccine is “very effective” for pregnant patients. An analysis of specialty-specific mean scores reveals a significant difference in opinions regarding the effectiveness of Td and TdaP vaccines (*p* = 0.012 and *p* < 0.001, respectively; Table 3).

Concerning the safety of the Td vaccine during pregnancy for both mothers and newborns, a substantial majority of physicians (78%) considered it “very safe”, as expected. Only 1.0% of PPs regarded it as “unsafe”. Most FPs and PPs expressed confidence in the safety of the influenza vaccine, with 50.9% and 53.3%, respectively, rating it “very safe”.

Most FPs (42.7%) noted that they were “uncertain” about the safety of the TdaP vaccine. However, all OGs regarded influenza, Td, and TdaP vaccines as either “safe” or “very safe”. Notably, the results indicate a significant difference in the mean scores concerning the safety of TdaP (*p* < 0.001; Table 4).

The majority of PPs (80%) and OGs (66%) indicated that they were aware of the disease burden of pertussis. In contrast, nearly half of FPs (44.7%) expressed uncertainty regarding this matter, demonstrating a statistically significant difference (*p* < 0.001; Table 5). 

In response to the question, “If you were pregnant, would the vaccination recommendation of your obstetrician and gynecologists be sufficient for your vaccination decision?”, most stated that a recommendation from their obstetrician would indeed be adequate for their vaccination decision (Table 6). 

In response to the question, “If you were pregnant, would the vaccination recommendation of your pediatricians following your other child be sufficient for your vaccination decision?”, 76.1% of PPs and 70.5% of FPs”indicated that they would follow the recommendation of a pediatrician (Table 7). 

In total, 67.7% of OG physicians reported sufficient knowledge about vaccination during pregnancy, surpassing all other physicians surveyed. These results highlight the need for increased awareness and knowledge regarding vaccination among FPs (74.5%) and PPs (77.2%) (Table 8).

According to OGs, 80% of expectant mothers consider vaccines to be “safe” or “extremely safe”, indicating a significant difference (*p* < 0.001; Table 9). The majority of OGs (58.9%) believed that pregnant women felt that there was insufficient medical information available to proceed with vaccination, whereas the majority of FPs and PPs expressed doubt regarding this claim, showing a significant variance (*p* < 0.001). Most FPs and PPs concurred that despite adequate medical evidence, pregnant women remained apprehensive about getting vaccinated (*p* < 0.001). 

Regarding pregnant women’s belief in the necessity of vaccines for their own and their babies’ protection, 40.6% of family practitioners agreed, while other specialties expressed uncertainty about this proposition, showing significant variance (*p* < 0.001). Over half of obstetricians and gynecologists (60.4%) disagreed with the view that postnatal vaccination could adequately protect infants according to pregnant women’s beliefs. Contrarily, some presented an opposing stance, signifying a substantial discrepancy (*p* < 0.001). Physicians, regardless of their specialty, agreed or strongly agreed that elucidating the implications of diseases and the advantages of vaccinating contributed to a higher acceptance rate for vaccines, indicating a considerable outcome (*p* = 0.001).

## 4. Discussion

Immunization is crucial during pregnancy to ensure the well-being of both mothers and their infants, both nationally and globally. Although the benefits of influenza and TdaP vaccines for pregnant women and their babies are well-known, the low utilization rate of these vaccines presents a significant obstacle from a preventive healthcare standpoint. Therefore, it is essential to understand the attitudes and perspectives of physicians toward vaccine acceptance to determine the existing levels of vaccine acceptance among pregnant women. In this regard, we have thoroughly examined these factors among physicians in Turkey. The outcomes of the questionnaires answered by the three specialties showed that 94.6% of the physicians who participated in this study recommended the Td vaccine. In contrast, 60.9% of physicians indicated that they would recommend the influenza vaccine during pregnancy, and only 45.5% of the physicians who responded to the survey recommended the TdaP vaccine. 

The influenza immunization rates, as defined by the outcomes of our study, while aligning with global statistics, still fall short of the desired rates. Our observations in this self-answered survey study reflect similar patterns in line with international recommendations. It is worth noting that in our study outcomes, the Td vaccine was the most effective among all vaccines, with an effectiveness rate of 84.8%, as rated by the responders. Despite having a low perception, the TdaP vaccine was deemed successful by 70.9% of all participating physicians. In contrast, the influenza vaccine had a success rate of 39.5%, yet it is recommended more frequently than the TdaP vaccine. Following the introduction of maternal pertussis immunization in the US in 2012, the incidence of pertussis in infants under six months dropped significantly. In 2014, the incidence was 169.0 per 100,000 population, while in 2018, it was 57.2 per 100,000. In addition, the mortality rate in infants below 12 months decreased from 16 per 100,000 in 2012 to 4 per 100,000 in 2018 [14]. A recent study by Skoff and colleagues also revealed that introducing the maternal TdaP vaccine led to a continuous decline in pertussis incidence in infants below two months of age [15]. This resulted in a reduction in the incidence gap between infants aged six months and those under 12 months. The study further indicated that maternal TdaP vaccination has a positive impact on the target age group by lessening the impact of pertussis. Furthermore, even preterm infants gained from maternal TdaP immunization. There was a correlation between an extended period between maternal vaccination and delivery, which resulted in elevated antibody levels in the cord blood of preterm babies, signifying the vaccine’s effectiveness.

From a safety perspective, 68.4% of all participating physicians in our study considered the TdaP vaccine either “safe” or “very safe”. However, the recommendation rate for the TdaP vaccine remained at 45.5%, despite 80.0% of PPs and 66.0% of OGs being aware of the pertussis disease burden. Similar trends are observed in developing countries. TdaP is widely used in almost all developed countries; however, its utilization in developing countries is still low [16]. Numerous physicians who responded to the study questionnaire acknowledge the importance of achieving successful outcomes with the TdaP vaccination. However, to effectively integrate it into their routine practice, they require more training and motivation from opinion leaders in the field, and direct recommendations from the government’s Ministry of Health will ensure more trust. Also, maternal immunization research needs to be improved so that local data collection and analysis may improve and identify coverage gaps and target interventions more effectively. Furthermore, it should be noted that economic factors may also contribute to the lower utilization rates. It is important to note that vaccinating pregnant women with TdaP is a highly cost-effective way to prevent diseases in infants who are not yet old enough to receive vaccinations. This method is also more effective than other strategies. However, TdaP is not currently recommended or reimbursed by the Turkish Ministry of Health guidelines. To prevent morbidity and mortality caused by pertussis, it is necessary to revise the guidelines and reimbursement strategies. By doing so, the adoption of TdaP vaccination among pregnant women can be enhanced, which will significantly contribute to disease prevention efforts. In our study, the physicians who responded to the questionnaire noted that many pregnant women are hesitant to get vaccinated due to their beliefs, knowledge, and attitudes, and 37.4% of physicians believe that pregnant women feel there is not enough medical data to support getting vaccinated, while 46.4% of physicians think that even with sufficient data, pregnant women are still afraid of getting vaccinated. This anxious behavior highlights the need for public awareness campaigns to address these concerns and encourage pregnant women to make informed decisions about their health. It is important to note that our study highlights the need for protection during pregnancy and until postnatal vaccination; however, as reported by the respondents, only 45.2% of pregnant women believe in protection during pregnancy, while 50.6% believe in protection until postnatal vaccination. This underscores the significance of education. The respondent physicians in our study unanimously agreed that explaining the risks of diseases and the benefits of vaccination increases vaccine acceptance among pregnant women. The necessity for education becomes evident, given that 74.5% of FPs and 77.2% of PPs express the need for more information regarding the effects and safety of vaccination programs during pregnancy. A crucial point is that since the knowledge of vaccines is increasing, updating knowledge is becoming critical. Continuous education and keeping healthcare professionals informed about the latest developments in vaccination safety are pivotal to ensuring the well-being of both pregnant women and infants.

According to our survey results, 66.1% of physicians agreed that only OG guidelines would be satisfactory for prenatal care. It is also noteworthy that this percentage increased to 72.5% if the same recommendation came from PPs. These findings suggest that patients consider immunization a duty of pediatricians, highlighting the critical role that pediatricians should play in leading immunization initiatives for expectant mothers. National programs may, therefore, consider vaccinating pregnant women with one dose of TdaP vaccine in the second or third trimester and at least 15 days before the end of pregnancy [17]. Based on the evidence, the Global Pertussis Initiative advocates maternal immunization during pregnancy as the primary strategy [18]. Similarly, The Advisory Committee on Immunization Practices also recommends that women receive a dose of TdaP during each pregnancy, preferably during the early part of gestation between 27 and 36 weeks [19,20]. These guidelines emphasize maternal immunization’s importance in safeguarding maternal and infant health against pertussis.

Vaccine acceptance always increases during the concomitant occurrence of infectious disease pandemics. During influenza outbreaks, since the risk of maternal morbidity and mortality is substantial, pregnant women always seek immunization as soon as the influenza vaccine becomes available. Similar principles apply during disease outbreaks. If our study was conducted during any of the infectious disease pandemics, like influenza, diphtheria, or pertussis, the results could be affected in a more positive way.

Despite high global vaccination coverage, pertussis epidemics continue to occur every 3–4 years, with variable trends across countries [16]. Therefore, it is crucial to ensure the safety of infants, children, adolescents, and adults through the optimal and prompt use of accessible vaccines. Maintaining vigilance and proactive vaccination efforts is essential to effectively prevent and control pertussis outbreaks. This study has certain limitations. Although in Turkey, there are 25,000 FPs, 11,000 PPs, and 6000 OGs, only 2387 of them responded to our survey (Table 1). This very low response rate is the most important limitation of our study.

## 5. Conclusions

Our research underscores the elements leading to insufficient vaccine adoption among healthcare workers in maternity and childcare. While OGs significantly influence maternal immunizations, PPs and FPs also play crucial roles. Since FPs are very well-established points of contact, our results show that support for FPs is very important. Therefore, regular training for healthcare providers on the latest vaccine recommendations and effective communication strategies to address vaccine hesitancy may reduce hesitancy. Our findings further stress the value of public awareness initiatives aimed at vaccinating expectant mothers. Visionary approaches, such as public vaccine education campaigns and improving professional knowledge about vaccine recommendations, could contribute to enhancing the vaccination schedule.

## Figures and Tables

**Table 1 vaccines-12-00798-t001:** Distribution of participating physicians by specialty.

	N	%
Family physicians	1296	54.3
Pediatricians	713	29.9
Obstetrician and gynecologists	378	15.8
Total	2387	100

**Table 2 vaccines-12-00798-t002:** Recommended vaccination during pregnancy.

		Not Recommended	Recommended in Case of Risk	Recommended Routinely	Total
Influenza	FPs, n (%)	61 (4.8%)	441 (35.0%)	757 (60.1%)	1259 (100%)
	PPs, n (%)	43 (6.1%)	228 (32.2%)	437 (61.7%)	708 (100%)
	OGs, n (%)	0 (0%)	138 (37.8%)	227 (62.2%)	365 (100%)
	Total, n (%)	104 (4.5%)	807 (34.6%)	1421 (60.9%)	2332 (100%)
Td	FPs, n (%)	12 (1.0%)	17 (1.4%)	1206 (97.7%)	1235 (100%)
	PPs, n (%)	31 (4.6%)	55 (8.2%)	587 (87.2%)	673 (100%)
	OGs, n (%)	0 (0%)	0 (0%)	366 (100%) *	366 (100%)
	Total, n (%)	43 (1.9%)	72 (3.2%)	2159 (94.9%)	2274 (100%)
Tdap	FPs, n (%)	567 (47.4%)	287 (24.0%)	342 (28.6%)	1196 (100%)
	PPs, n (%)	163 (23.1%)	132 (18.7%)	411 (58.2%)	706 (100%)
	OGs, n (%)	0 (0%)	85 (23.4%)	278 (76.6%) *	363 (100%)
	Total, n (%)	730 (32.2%)	504 (22.3%)	1031 (45.5%) *	2265 (100%)

FPs: family physicians; PPs: pediatricians; OGs: obstetricians and gynecologists; Td: tetanus and diphtheria vaccine; Tdap: tetanus, diphtheria, and pertussis vaccine; Statistical test: Pearson’s chi-square test; Significance: *p* < 0.001 *.

**Table 3 vaccines-12-00798-t003:** Effectiveness of vaccination during pregnancy to protect mother and newborn against infectious diseases.

		Mean (SD)	Min–Max	*p*-Value
Influenza	FPs	4.29 (0.78)	2–5	0.219 ^a^
	PPs	4.22 (0.87)	1–5	
Td	FPs	4.82 (0.44)	3–5	0.012 ^b^
	PPs	4.82 (0.58)	1–5	
	OGs	4.80 (0.40)	4–5	
Tdap	FPs	3.94 (0.96)	1–5	<0.001 ^b^
	PPs	4.41 (0.95)	1–5	
	OGs	4.76 (0.43)	4–5	

FPs: family physicians; PPs: pediatricians; OGs: obstetricians and gynecologists; Td: tetanus and diphtheria vaccine; Tdap: tetanus, diphtheria, and pertussis vaccine; SD: standard deviation; min: minimum; max: maximum. Out of all OG physicians (n = 365), only 4 voted for influenza vaccine; therefore, this sub-group was not included in the statistical analysis. Statistical tests: ^a^: Mann–Whitney U test; ^b^: Kruskal–Wallis test.

**Table 4 vaccines-12-00798-t004:** Safety of vaccination during pregnancy for mother and newborn.

		Mean (SD)	Min–Max	*p*-Value ^a^
Influenza	FPs	4.30 (0.82)	2–5	0.023
	PPs	4.24 (0.98)	1–5	
	OGs	4.50 (0.50)	4–5	
Td	FPs	4.78 (0.45)	3–5	0.031
	PPs	4.68 (0.66)	1–5	
	OGs	4.75 (0.43)	4–5	
Tdap	FPs	3.89 (0.99)	1–5	<0.001
	PPs	4.32 (0.90)	1–5	
	OGs	4.76 (0.43)	4–5	

FPs: family physicians; PPs: pediatricians; OGs: obstetricians and gynecologists; Td: tetanus and diphtheria vaccine; Tdap: tetanus, diphtheria, and pertussis vaccine; SD: standard deviation; min: minimum; max: maximum. Statistical tests: ^a^: Kruskal–Wallis test.

**Table 5 vaccines-12-00798-t005:** Knowledge of physicians about the disease burden of pertussis.

	No	Yes	Uncertain	Total
FPs, n (%)	213 (17.0%)	481 (38.3%)	562 (44.7%)	1256 (100%)
PPs, n (%)	3 (4%)	560 (80.0%) **	137 (19.6%)	700 (100%)
OGs, n (%)	0 (0%)	237 (66.6%) **	119 (33.4%)	356 (100%)
Total, n (%)	216 (9.3%)	1278 (55.3%)	818 (35.4%)	2312 (100%)

FPs: family physicians; PPs: pediatricians; OGs: obstetricians and gynecologists. Statistical tests: Pearson’s chi-square test. Significance: *p* < 0.001 **.

**Table 6 vaccines-12-00798-t006:** If you were pregnant, would the vaccination recommendation of your obstetrician and gynecologists be sufficient for your vaccination decision?

	Physician’s Recommendation Would Be Sufficient	I Would Decide after Searching the Issue	Total
FPs, n (%)	788 (63.1%)	461 (36.9%)	1249 (100%)
PPs, n (%)	443 (63.6%)	253 (36.4%)	696 (100%)
OGs, n (%)	288 (81.6%) **	65 (18.4%)	353 (100%)
Total, n (%)	1519 (66.1%)	779 (33.9%)	2298 (100%)

FPs: family physicians; PPs: pediatricians; OGs: obstetricians and gynecologists. Statistical test: Pearson’s chi-square test; Significance: *p* < 0.001 **.

**Table 7 vaccines-12-00798-t007:** If you were pregnant, would the vaccination recommendation of your pediatricians following your other child be sufficient for your vaccination decision?

	Physician’s Recommendation Would Be Sufficient	I Would Decide after Searching the Issue	Total
FPs, n (%)	879 (70.5%)	367 (29.5%)	1246 (100%)
PPs, n (%)	524 (76.1%) *	165 (23.9%)	689 (100%)
Total, n (%)	1403 (72.5%)	532 (27.5%)	1935 (100%)

FPs: family physicians; PPs: pediatricians. Statistical test: Pearson’s chi-square test. Significance: *p* < 0.009 *.

**Table 8 vaccines-12-00798-t008:** Do you need detailed information about vaccination during pregnancy?

	No	Yes	Total
FPs, n (%)	317 (25.5%)	927 (74.5%)	1244 (100%)
PPs, n (%)	157 (22.8%)	531 (77.2%)	688 (100%)
OGs, n (%)	239 (67.7%) **	114 (32.3%)	353 (100%)
Total, n (%)	713 (100%)	1572 (100%)	2285 (100%)

FPs: family physicians; PPs: pediatricians; OGs: obstetricians and gynecologists. Statistical test: Pearson’s chi-square test. Significance: *p* < 0.001 **.

**Table 9 vaccines-12-00798-t009:** Vaccine acceptance by pregnant women.

		Mean (SD)	Min–Max	*p*-Value ^a^
**Q1:** Pregnant women generally think that vaccines are safe.	FPs	3.49 (0.92)	2–5	<0.001
	PPs	3.39 (1.20)	1–5	
	OGs	4.00 (0.63)	3–5	
**Q2:** Pregnant women think that there is not enough medical data to receive the vaccine.	FPs	2.94 (1.07)	1–5	<0.001
	PPs	3.33 (1.05)	1–5	
	OGs	3.59 (0.49)	3–4	
**Q3:** Pregnant women are afraid to receive any vaccine, even if there is sufficient medical data.	FPs	3.24 (1.22)	1–5	<0.001
	PPs	3.56 (1.11)	1–5	
	OGs	2.81 (0.75)	2–4	
**Q4:** Pregnant women think that the vaccine is necessary to protect their babies and themselves.	FPs	3.66 (0.89)	2–5	<0.001
	PPs	3.09 (0.99)	1–5	
	OGs	2.80 (0.74)	2–4	
**Q5:** Pregnant women think protecting their babies with postnatal vaccination is sufficient.	FPs	3.52 (1.10)	1–5	<0.001
	PPs	3.55 (1.09)	1–5	
	OGs	2.60 (0.80)	2–4	
**Q6:** Explaining the disease risks and benefits of vaccination to pregnant women increases the acceptance rate of the vaccine.	FPs	4.39 (0.76)	1–5	0.001
	PPs	4.27 (0.81)	1–5	
	OGs	4.40 (0.80)	3–5	

FPs: family physicians; PPs: pediatricians; OGs: obstetricians and gynecologists; SD: standard deviation; min: minimum; max: maximum. Statistical test: ^a^: Kruskal–Wallis test.

## Data Availability

The data presented in this study are available on request from the corresponding author due to privacy

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
