# Peer review of "Comparison of Physicians’ Attitudes and Practice Regarding Vaccination during Pregnancy in Turkey"

_vaccines, 2024, doi:10.3390/vaccines12070798_

Round 1

Reviewer 1 Report

Comments and Suggestions for Authors

This is an interesting work, in which authors expose that although there are many guidelines for vaccinating pregnants, there are concerns about which vaccines should be administered during pregnancy. They justify that one important point for low immunization rates is economic burden. Authors investigated 3 groups of physicians.

It is not clear if all physicians are from the same hospitalar unit. In that case, how representative the number of participants could be in a hole?

Abstract, line 26, please add the meaning of TdaP.

Table 3, I did not understand why OG does not appear in line of Influenza.

Taking into account the results, what do authors could suggest in order to improve the immunization protocols and promote vaccines , mainly the ones that received low numbers of recommendations?   

Author Response

Thank you very much for your review and for your valuable comments. The points that you raised were added to our manuscript.

This is an interesting work, in which authors expose that although there are many guidelines for vaccinating pregnants, there are concerns about which vaccines should be administered during pregnancy. They justify that one important point for low immunization rates is economic burden.

1) Authors investigated 3 groups of physicians. It is not clear if all physicians are from the same hospitalar unit. In that case, how representative the number of participants could be in a whole?

This sentence is now revised. As the study was conducted and completed during the initial phase of the COVID-19 pandemic, only 2387 physicians responded to the questionnaire from different hospitals and various cities of Turkey . Line 84-85

2) Abstract, line 26, please add the meaning of TdaP.

This was added as tetanus, diphtheria and pertussis vaccine. Line 26

3) Table 3, I did not understand why OG does not appear in line of Influenza.

Below sentence was added to Table 3

As all OG physicians (n=365) voted as  4 only for influenza vaccine,  this sub-group was not included into the statistical analysis Line 142

4) Taking into account the results, what do authors could suggest in order to improve the immunization protocols and promote vaccines, mainly the ones that received low numbers of recommendations?  

This was added to discussion line 242-245 as follows:

In order to successfully incorporate this into their regular practice, healthcare professionals need additional training and encouragement. This should be provided not only by key opinion leaders in the field, but also through direct recommendations from the Ministry of Health where endorsements from the government will help to build more trust. Furthermore, there is a need to enhance research on maternal immunization. Local data collection and analysis could help to identify coverage gaps and target interventions more effectively.

Reviewer 2 Report

Comments and Suggestions for Authors

The paper is very interesting because it is necessary to know the attitudes of healthcare professionals in relation to vaccines in general and during pregnancy in particular. I would like to make some comments. In the data analysis section it is said that a logistic regression was performed but it is not said what the dependent variable of this regression is. Nor is it said why this analysis was done. Additionally, no results from this analysis are shown. In the tables where the Chi Square test is used, nothing appears that refers to the p-value. Please put the p – value or something that indicates it.

Also, in the case of standard deviation it is convenient to remove the ± sign that accompanies the standard deviation. The standard deviation is a measure of dispersion and not of precision. I offer two options: put Mean ± Standard Error or mean (sd).

- Altman DG, Gore SM, Gadner MJ, Pocock SJ. Statistical guidelines for contributors to medical journals. Br Med J 1983;286: 1,489-1,493.

- Bailar JC, Mosteller F. Guidelines for statistical reporting in articles for medical journals: amplifications and explanations. Ann Intern Med 1988;108: 266-273.

- Tobias A. [Mean +/- SD, an incorrect expression].Med Clin (Barc). 1998 Feb 7;110(4):157

Author Response

Thank you very much for your review and for your valuable comments. The points that you raised were added to our manuscript.

The paper is very interesting because it is necessary to know the attitudes of healthcare professionals in relation to vaccines in general and during pregnancy in particular.

I would like to make some comments.

1) In the data analysis section it is said that a logistic regression was performed but it is not said what the dependent variable of this regression is. Nor is it said why this analysis was done.

Additionally, no results from this analysis are shown.

This must be an editorial mistake. Thanks for referring to it. No logistical regression methods were uses during the statistical analysis of the study.

The sentence should read as follows:

Statistical Analysis: Pearson Chi-square test was used to compare categorical variables, and Mann Whitney-U for two independent samples and Kruskal Wallis for multiple independent samples to determine the factors predicting vaccine recommendations during pregnancy. Statistical significance was defined as p<0.05. Missing values, which were not completed through queries were not extrapolated. Line 107-112

2) In the tables where the Chi Square test is used, nothing appears that refers to the p-value. Please put the p – value or something that indicates it. 

Also, in the case of standard deviation it is convenient to remove the ± sign that accompanies the standard deviation.

The standard deviation is a measure of dispersion and not of precision. I offer two options: put Mean ± Standard Error or mean (sd).

- Altman DG, Gore SM, Gadner MJ, Pocock SJ. Statistical guidelines for contributors to medical journals. Br Med J 1983;286: 1,489-1,493.

- Bailar JC, Mosteller F. Guidelines for statistical reporting in articles for medical journals: amplifications and explanations. Ann Intern Med 1988;108: 266-273.

- Tobias A. [Mean +/- SD, an incorrect expression].Med Clin (Barc). 1998 Feb 7;110(4):157

Performed as noted, and the article by Altman et.al, was taken as the main reference. Tables 3,4 and 9.

Reviewer 3 Report

Comments and Suggestions for Authors

The manuscript is generally well written and scientific sounds. There are some general statements that could be revised. For example:

 “…During pregnancy and early infancy, sex hormones may alter the immune system response, leaving mothers and infants vulnerable to infections …” In fact, sex hormones interact with potential immune responses all along the life, besides, I would not consider that they induce vulnerability to infections (even considering the occurrence of immunossupressive mechanisms during pregnancy).

“These antibodies protect infants from birth until the…” Probably authors want to refer to antibodies induced by vaccination during pregnancy, but they should better state their point at the manuscript, since no “antibodies” were previously mentioned.

“Additionally, the economic burden on an expectant mother and her family is one of the primary reasons for low immunization rates.” There are countries were vaccination is provided in the context of social/public policies. Maybe this point should be highlighted.

Please, specify at 3.2. Physician Perceptions “The most commonly recommended vaccine was Td, endorsed by 94.9% of all participating physicians.” In fact, according to Table 2, 94.9% recommended routinely Td vaccination plus 3.2% recommended in case of risk, which is still a recommendation”

Considering “The very low response rate is the most important limitation of the study.” I wonder what would be the total number of physicians eligible to this survey in Turkey? At least an estimated number could be presented.

I believe that concerns about knowledge of family physicians should be highlighted as the main point and conclusion arousing from this survey. Please, put more emphasis in your conclusion about the needs for “public vaccine education campaigns and improving professional knowledge about vaccine recommendations”

Finally, considering the fact that “The study was conducted and completed during the initial phase of the COVID-19 pandemic…” I wonder if the results could be influenced by the concomitant occurrence of an infectious disease pandemics. This point deserves to be discussed.

Minor point

At all Tables, standardize 100% (instead of 100.0%)

Author Response

Thank you very much for your review and for your valuable comments. The points that you raised were added to our manuscript.

The manuscript is generally well written and scientific sounds. There are some general statements that could be revised. For example:

1) “…During pregnancy and early infancy, sex hormones may alter the immune system response, leaving mothers and infants vulnerable to infections …” In fact, sex hormones interact with potential immune responses all along the life, besides, I would not consider that they induce vulnerability to infections (even considering the occurrence of immunossupressive mechanisms during pregnancy).

That sentence was changed

“Hormonal changes taking place during During pregnancy and early infancy, sex hor-mones may alter the immune system response, leaving mothers and infants vulnerable to infections that can be prevented by immunization” Line 34-35-36

2) “These antibodies protect infants from birth until the…” Probably authors want to refer to antibodies induced by vaccination during pregnancy, but they should better state their point at the manuscript, since no “antibodies” were previously mentioned.

That sentence was changed

Maternal vaccines have the potential to offer significant protection for both mothers and infants. This is achieved through the production of antibodies with vaccination during pregnancy. These antibodies safeguard infants from birth until they receive their primary immunization series, providing crucial protection for newborns against diseases such as tetanus and whooping cough, which can be life-threatening [1, 2]. Lines 43-48

3) “Additionally, the economic burden on an expectant mother and her family is one of the primary reasons for low immunization rates.” There are countries were vaccination is provided in the context of social/public policies. Maybe this point should be highlighted.

This was added to text line 62-65

Additionally, the economic burden on an expectant mother and her family is one of the primary reasons for low immunization rates. Although NIS data found that main reason for not vaccinated was vaccine safety or efficacy (44.8%) and cost was only (5.6%). (Smith PJ, Humiston SG, Parnell T, Vannice KS, Salmon DA. The association between intentional delay of vaccine administration and timely childhood vaccination coverage. Public Health Rep 2010;125(4):534–41.) Line 62-65   

4) Please, specify at 3.2. Physician Perceptions “The most commonly recommended vaccine was Td, endorsed by 94.9% of all participating physicians.” In fact, according to Table 2, 94.9% recommended routinely Td vaccination plus 3.2% recommended in case of risk, which is still a recommendation”

That sentence was changed

The most commonly recommended vaccine by all participating physicians was Td (94.9% routinely recommended, and an additional 3.2% recommended in case of risk).  Lines 120-122

5) Considering “The very low response rate is the most important limitation of the study.” I wonder what the total number of physicians would be eligible to this survey in Turkey? At least an estimated number could be presented.

This was added to discussion

Although in Turkey there are 25.000 FP, 11.000 PP and 6.000 OG only 2387 of them responded our survey. (Table 1) This very low response rate is the most important limitation of the study. Line 289-291

6) I believe that concerns about knowledge of family physicians should be highlighted as the main point and conclusion arousing from this survey. Please, put more emphasis in your conclusion about the needs for “public vaccine education campaigns and improving professional knowledge about vaccine recommendations”

This was added to conclusion Line 303-306

Since FP are very well-established point of contact, our results show that support of FP is very important. Therefore, regular training for healthcare providers on the latest vaccine recommendations and effective communication strategies to address vaccine hesitancy may reduce hesitancy. Line 303-306

7) Finally, considering the fact that “The study was conducted and completed during the initial phase of the COVID-19 pandemic…” I wonder if the results could be influenced by the concomitant occurrence of an infectious disease pandemics. This point deserves to be discussed.

This was incorporated to the discussion line 291-296

Vaccine acceptance always increases during concomitant occurrence of an infec-tious disease pandemics. During influenza outbreaks since the risk of maternal morbid-ity and mortality is substantial, pregnant women always seeks for immunization as soon as influenza vaccine becomes available. Similar principles apply during disease outbreaks. If our study were conducted during any of the infectious disease pandemics like influenza, diphtheria or pertussis results could be affected in more positive way Line 291-296

Minor point

1) At all Tables, standardize 100% (instead of 100.0%)

That was corrected throughout the article.

Round 2

Reviewer 1 Report

Comments and Suggestions for Authors

In my opinion, all my questions and observations have been adequately answered.

Author Response

Thank you very much 

regards

Ates

Reviewer 2 Report

Comments and Suggestions for Authors

Dear Author, Thank you very much for answering and clarifying all my doubts. I make a comment, in the document that I have received with the corrections the logistic regression models continue to appear and in table 9 there is still the +- sign next to the standard deviation. Please review.

Author Response

Thank you very much for your review. Although we answered you in the letter, we forgot to add our correction to the text. This mistake has been corrected, and the revised manuscript is ready for your review.